# Monitoring Spatiotemporal Vegetation Response to Drought Using Remote Sensing Data

**DOI:** 10.3390/s23042134

**Published:** 2023-02-14

**Authors:** Salman Mirzaee, Ali Mirzakhani Nafchi

**Affiliations:** 1Department of Agronomy, Horticulture and Plant Science, College of Agriculture, Food and Environmental Sciences, South Dakota State University, Brookings, SD 57007, USA; 2Extension Precision Agriculture Specialist, College of Agriculture, Food and Environmental Sciences, South Dakota State University, Brookings, SD 57007, USA

**Keywords:** precision agriculture, aspatial, spatial, Normalized Difference Vegetation Index, Normalized Multiband Drought Index, spectral indices

## Abstract

Environmental factors such as drought significantly influence vegetation growth, coverage, and ecosystem functions. Hence, monitoring the spatiotemporal vegetation responses to drought in a high temporal and adequate spatial resolution is essential, mainly at the local scale. This study was conducted to investigate the aspatial and spatial relationships between vegetation growth status and drought in the southeastern South Dakota, USA. For this purpose, Landsat 8 OLI images from the months of April through September for the years 2016–2021, with cloud cover of less than 10%, were acquired. After that, radiometric calibration and atmospheric correction were performed on all of the images. Some spectral indices were calculated using the Band Math toolbox in ENVI 5.3 (Environment for Visualizing Images v. 5.3). In the present study, the extracted spectral indices from Landsat 8 OLI images were the Normalized Difference Vegetation Index (NDVI) and the Normalized Multiband Drought Index (NMDI). The results showed that the NDVI values for the month of July in different years were at maximum value at mostly pixels. Based on the statistical criteria, the best regression models for explaining the relationship between NDVI and NMDI_Soil_ were polynomial order 2 for 2016 to 2019 and linear for 2021. The developed regression models accounted for 96.7, 95.7, 96.2, 88.4, and 32.2% of vegetation changes for 2016, 2017, 2018, 2019, and 2021, respectively. However, there was no defined trend between NDVI and NMDI_Soil_ observed in 2020. In addition, pixel-by-pixel analyses showed that drought significantly impacted vegetation coverage, and 69.6% of the pixels were negatively correlated with the NDVI. It was concluded that the Landsat satellite images have potential information for studying the relationships between vegetation growth status and drought, which is the primary step in site-specific management.

## 1. Introduction

The limitations of land resources for agricultural production in the world have created a deep concern for decision-makers. Environmental challenges, especially land-cover and climate changes, are the main resource limitations caused by the gradual degradation via human activity [1,2,3]. For the best management of agricultural land resources, it is important to consider the environmental factors and understand some limitations of the productivity of agricultural lands.

Vegetation coverage and drought play important roles in making conditions favorable or unfavorable for agriculture production. Remote sensing is a widely applied information-gathering tool with a minimal cost for studying scientific evidence of changes in environmental factors, especially in agriculture [4,5]. Different extracted spectral indices from satellite images have been employed to monitor and detect the effect of environmental challenges, including climate change effects [6,7], land degradation [2], forest fire detection [8], and so on. The Normalized Difference Vegetation Index (NDVI) [9] is by far the most commonly used type of spectral index for remote sensing, which applies the information of NIR and Red bands from a satellite image to detect vegetation coverage in the field. This could be related to the close relationship between the physiological characteristics of plants and NDVI [10]. NDVI not only shows the changing characteristics and distribution of vegetation coverage, but it also can provide valuable environmental information, such as that regarding the effects of climate change [11]. In this way, in several studies, NDVI was applied to explain moisture, nitrogen, growth stage [10], soil properties [12,13,14], and climate change [7,15,16,17].

Drought is a serious environmental problem with the greatest economic and social impact in the world [18]. The Normalized Multiband Drought Index (NMDI) is applied to detect vegetation and soil-water content using satellite images [8,19,20]. Wang and Qu reported that using the combination of reflectance in the 0.86, 1.64, and 2.13 µm bands could estimate soil moisture and offer more accurate assessments of drought severity [19]. Most studies have focused on employing NMDI to predict and evaluate water status in forest conditions [8,20].

Understanding the spatiotemporal vegetation response to drought at a high temporal and adequate spatial resolution is necessary to ensure the safety of lives and property [21,22]. However, only a few similar studies have focused on studying distribution patterns between spectral vegetation and drought indices on a local spatial scale using remote sensing data. Studying a local spatial scale is important to reduce the impact of geomorphology and topography. The main objectives of the present study were to: (i) monitor and detect vegetation coverage and drought by using spectral indices extracted from remote sensing data (i.e., Landsat 8 OLI images), (ii) model the vegetation response to drought by using spectral indices, and (iii) investigate the spatial variability of vegetation in conjunction with drought.

## 2. Material and Methods

### 2.1. Study Area

The general landscape of the study area (43°18′20″ to 43°18′46″ N and 96°54′56″ to 96°54′20″ E) is located in the southeastern part of the State of South Dakota, USA (Figure 1). The studied region covers a 56.4 ha area of Lincoln County. According to Figure 1, the elevation from sea level at Lincoln County varies from 357 and 478 m. This area is plowed by a moldboard at a depth of 20–30 cm for the rain-fed agriculture production of such crops as soybeans, corn, and wheat in rotation. Generally, drought is the main challenge in this region.

### 2.2. Climate Factors

The precipitation and temperature data from 2016 to 2021 for the studied region were obtained from the Meteorological Data Service Center (https://prism.oregonstate.edu/explorer/ (accessed on 9 November 2022)). The monthly temperature and precipitation are presented for the studied area for the period from 2016 to 2021 in Figure 2. More than 80% of precipitation occurs in the period from March to October (Figure 2a). The cumulative precipitation varied from 455.2 to 1104.9 mm during the period of 2016–2021 (Figure 2a). The distribution of precipitation in this area is uneven. In addition, based on Figure 2b, the annual mean temperature ranged from 7.05 to 9.58 °C during the period of 2016–2021 for the studied region. This area is classified as having a very humid climate.

### 2.3. Satellite Image Processing

The Landsat 8 Level-1 images for the months of April–September were acquired for the years 2016–2021 from the National Aeronautics and Space Administration Agency (NASA) server. The path/row was 29/30. The cloud cover in the applied images was less than 10%. The OLI sensor onboard Landsat 8 captures data in several spectral bands, with a resolution of 30 m, as presented in Table 1.

#### 2.3.1. Image Preprocessing

First, radiometric calibration was performed by using the radiance calibrating method in the Radiometric Calibration toolbox, and the atmospheric correction was achieved by using a quick method in the QUAC toolbox in ENVI 5.3 (Environment for Visualizing Images 5.3). Then, the reflectance was calculated by the digital numbers of the bands ×0.0001 in the Band Math toolbox in ENVI 5.3. The spectral indices, such as the Normalized Difference Vegetation Index (NDVI) and the Normalized Multiband Drought Index (NMDI) were calculated using the Band Math toolbox in ENVI 5.3.

#### 2.3.2. Vegetation Coverage Index

NDVI, an important spectral vegetation index, was employed to remote-sense vegetation coverage based on Equation (1) [9,23]:(1)$$\mathrm{NDVI}=\frac{B5-B4}{B5+B4}$$

For calculating this spectral index, B4 and B5, i.e., red and near-infrared reflectance (defined in Table 1), respectively, were used. The negative values of this spectral index (from −1 to zero) indicate water resources; a zero-value represents no green leaves, 0.2 to 0.3 values show the bare soil, and higher values (close to +1) represent the highest possible density of vegetation coverage.

#### 2.3.3. Drought Index

NMDI (Normalized Multiband Drought Index), introduced by Wang and Qu [19], uses the reflectance information of the near-infrared and short-wave infrared bands to monitor vegetation water contents. Wang et al. [8] modified the NMDI to monitor soil moisture conditions with Equation (2):(2)$${\mathrm{NMDI}}_{\mathrm{Soil}}=0.9-\frac{R_{0.86\;\mu m}-\left(R_{1.64\;\mu m}-R_{2.13\;\mu m}\right)}{R_{0.86\;\mu m}+\left(R_{1.64\;\mu m}-R_{2.13\;\mu m}\right)}$$

Landsat OLI reflectance bands 5, 6, and 7 (i.e., Near-infrared, Short-wave infrared reflectance 1 and Short-wave infrared reflectance 1, defined in Table 1, respectively) were employed for $R_{0.86\;\mu m}$, $R_{1.64\;\mu m}$, and $R_{2.13\;\mu m}$ to calculate NMDI_Soil_ (Equation (2)). The NMDI_Soil_ values range from 0 to 0.9, with the higher values representing increasing soil drought. Values less than 0.3 show extremely wet soil, 0.3–0.5 indicate intermediate soil moisture contents, 0.5–0.7 show dry soil, and 0.7–0.9 show very dry, bare soil [8]. ArcGIS 10.3 was applied to create the thematic maps.

### 2.4. Statistical Analysis

Regression models, such as simple linear and polynomial, were used to link the vegetation and drought data. The regression models are described as:(1)Simple linear regression:



(3)
$$y_{i}=\beta_{0}+\beta_{1}x_{i}+\varepsilon_{i}$$

(2)Polynomial regression:


(4)$$y_{i}=\beta_{0}+\beta_{1}x_{i}+\beta_{2}x_{i}{}^{2}+\cdots +\beta_{p}x_{i}{}^{p}+\varepsilon_{i}$$
where $y_{i}$ is a dependent variable; $x_{i}$ is an independent variable; $\varepsilon_{i}$ is the error; and $\beta_{0}$, $\beta_{1}$, …, $\beta_{p}$ are regression coefficients. In the present research, the cross-validation approach was applied to compare the accuracy of different, developed regression models by using ME (mean error, Equation (5)), RMSE (root-mean-square error, Equation (6)), R^2^ (coefficient of determination, Equation (7)) and r (Pearson correlation, Equation (8)):

(5)$$\mathrm{ME}=\frac{1}{N}\sum_{i=1}^{N}(P_{i}-M_{i})$$(6)$$\mathrm{RMSE}={\left[\frac{\sum_{i=1}^{N}{\left(P_{i}-M_{i}\right)}^{2}}{N}\right]}^{0.5}$$(7)$$R^{2}=1-\frac{\sum_{i=1}^{N}{\left(M_{i}-P_{i}\right)}^{2}}{\sum_{i=1}^{N}{\left(M_{i}-\bar{M_{1}}\right)}^{2}}\;$$(8)$$r=\frac{\sum_{i=1}^{N}(P_{i}-\hat{P})\left(M_{i}-\bar{M}\right)}{\sqrt{\sum_{i=1}^{N}{\left(P_{i}-\bar{P_{1}}\right)}^{2}\sum_{i=1}^{N}{\left(M_{i}-\bar{M_{1}}\right)}^{2}}}$$
where N is the number of data, $P_{i}$ shows the estimated data, $\bar{P_{1}}$ is the mean of the predicted data, $M_{i}$ represents the measured data, and $\bar{M_{1}}$ indicates the mean of the measured data. The F-test was applied to check the significance of the correlation coefficient between different environmental factors. Nonsignificant was applied to values where *p* > 0.1, weakly significant was applied to values where *p* < 0.1, moderately significant was applied to values where *p* < 0.05, and highly significant was applied to values where *p* < 0.01.

## 3. Results and Discussion

### 3.1. Spectral Characteristics

Some spectral curves for randomly selected pixels from the month of July, from 2016 to 2021, are presented in Figure 3. Overall, the reflectance values varied a lot for each year (Figure 3). According to the reflectance values in Figure 3, the higher reflectance variation was for the B5 (NIR band) wavelength range (Figure 3).

### 3.2. Results of Vegetation Monitoring by Using NDVI

In the present study, vegetation coverage variations were detected using NDVI calculated from Landsat 8 OLI from 2016 to 2021. The NDVI values from April to August in the study period against time and cumulative precipitation are presented in Figure 4. As can be seen from Figure 4a, NDVI values for all pixels increased with increasing cumulative precipitation. Precipitation, as the main climate factor, and NDVI were applied to investigate the effect of climate change on the vegetation coverage [1,7]. Our results (Figure 4a) showed a general trend in increasing NDVI with increasing precipitation. However, our results differed widely for some years (Figure 4a). This could be related to the complex relationships between precipitation and vegetation characteristics [17], human activity [24], and the research scale [25].

The analysis of NDVI distributions over time is presented in Figure 4b. As can be seen from Figure 4b, the NDVI values for the month of July for different years, by soybean, corn, and wheat in rotation, at most pixels were at the maximum. For this reason, the NDVI values for the month of July were analyzed in the present study. The descriptive statistics for NDVI values for the month of July in different years are presented in Table 2 and Figure 5 for all pixels. The lowest and highest mean NDVI values were 0.757 and 0.862 for the month of July in 2017 and 2018, respectively (Table 2). The minimum and maximum NDVI values ranged from 0.276 to 0.531, and from 0.887 to 0.956, respectively (Table 2, Figure 5). The minimum (7.20%) and maximum (18.73%) CV values were found for 2021 and 2020, respectively (Table 2, Figure 5).

The spatial pattern of NDVI values for the month of July from 2016 to 2021, by soybean, corn, and wheat in rotation, is presented in Figure 6. As seen from Figure 6a,f, the NDVI values were almost constant for many pixels in 2016 and 2021. By contrast, the NDVI values were nonuniform for 2017 and 2020. The lowest NDVI values in 2017 and 2020 were observed in the southwest and the northwest of the studied area, respectively, for many pixels (Figure 6b,e).

In the present study, the spatial patterns of NDVI changes in the months of July for 2016–2017, 2017–2018, 2018–2019, 2019–2020, and 2020–2021 are shown in Figure 7. The area percentages of each class of NDVI changes are presented in Table 3. From 2016 to 2017, the NDVI values decreased and increased to 73.86 and 26.14% of the total area, respectively (Table 3). As can be seen from Figure 7a, the biggest reduction in NDVI values occurred in the southern part in comparison with the eastern part of the studied area. By contrast, from 2017 to 2018, the NDVI values decreased and increased in 1.89 and 98.1% of the studied area, respectively (Table 3). Figure 7b shows that the NDVI values from 2017 to 2018 mostly increased in all parts of the studied area, especially in the southern part. The NDVI values from 2018 to 2019 decreased and increased by 78.11 and 21.89% of the total area, respectively (Table 3). The NDVI values mostly changed in the southern part of the studied area (Figure 7c). Figure 7d,e shows that the biggest changes from 2019 to 2020 and from 2020 to 2021 mostly occurred in the northwestern part of the studied area. In addition, small changes can be observed in other parts of the studied area from 2019 to 2020 and from 2020 to 2021 (Figure 7d,e).

### 3.3. Results of Drought Monitoring by Using NMDI_Soil_

Figure 8 presents the NMDI_Soil_ values against the times for the months of April through August for the years 2016–2021 for all pixels in the studied area. Based on Figure 8a, the values of NMDI_Soil_ at the beginning of the growing season were higher (>0.5) for 2016, 2017, and 2018, and then they decreased with time (i.e., increasing soil moisture content). By contrast, this pattern was the opposite for 2020 and 2021 (i.e., decreasing soil moisture with time) (Figure 8a). The soil moisture content decreases with the increasing NMDI_Soil_ value [19]. On the basis of the results in Table 4, the CV values for the month of July ranged from 8.11% (2021) to 19.38% (2019) (Table 4).

Figure 9 shows the spatial pattern of drought trends for the month of July, from 2016 to 2021. In general, the NDMI_Soil_ values increased by many pixels from 2016 to 2021 (Figure 9). Increasing NMDI_Soil_ values show the soil moisture content decreased from 2016 to 2021 in many pixels (Figure 9).

### 3.4. Monitoring Spatiotemporal Vegetation Response to Drought

#### 3.4.1. The Aspatial Relationships

For the present study, Figure 10 gives the scatter plot of NDVI and NMDI_Soil_ in the study period for the month of July in each year. The Pearson correlation coefficient between NDVI and NMDI_Soil_ varied from 0.174 to 0.977 in the studied period (Figure 10). The results show that the best regression model for explaining the relationship between NDVI and NMDI_Soil_ is polynomial order 2, based on the ME, R^2^, and RMSE criteria from 2016 to 2019 (Table 5). In addition, the results in Table 5 indicate that the R^2^ value ranged from 0.322 (2021) to 0.967 (2016). The analysis of the ME criterion showed that the developed regression models underestimated the NDVI values (−0.0001 to −0.0004), except in 2021 (Table 5). According to the presented results in Table 5, the RMSE values varied from 0.014 to 0.048. The smallest RMSE was achieved in 2016 (Table 5). However, there was no defined trend between NDVI and NMDI_Soil_ in 2020 (Figure 10). This could be an effect of the amount of cumulative precipitation (455.2 mm) in 2020 in comparison with 2016 (849.2 mm), 2017 (809.2 mm), 2018 (1015.6 mm), 2019 (1104.9 mm), and 2021 (823.6 mm) (Figure 2a).

#### 3.4.2. The Spatiotemporal Analysis

The vegetation response to drought could be illustrated quantitatively through an analysis of the correlation between spectral vegetation and drought indices. Figure 11a shows the results of pixel-by-pixel correlation analysis between NMDI_Soil_ and NDVI from 2016 to 2021. The results indicated that the pixel-by-pixel correlation ranged from −0.91 to 0.89 in the studied region (Figure 11a). There was either a negative or a positive correlation between NDVI and NMDI_Soil_, which indicated an increasing or decreasing trend of vegetation response to drought. From the correlation values, the percentage of pixels negatively correlated between NMDI_Soil_ and NDVI was 69.4% of the total pixels for all of the studied years (Figure 11b), of which 0.22, 2.94, and 7.21% correlated as extremely (*p* < 0.01), moderately (*p* < 0.05), or weakly (*p* < 0.1) significant, respectively. According to Figure 11a, the high vegetation response to drought occurred in the southern and northern parts of the studied area. The low correlation of vegetation coverage to drought was mainly distributed in the eastern parts of the studied area (Figure 11a). However, in the northwestern parts of the studied area, vegetation increased with increasing drought (Figure 11a). This could be due to the complicated nature of the field, for instance, the effects of soil texture and organic matter.

Further analyses were performed on the spatial variation between NMDI_Soil_ and NDVI for the highly correlated years (i.e., 2016, 2017, 2018, and 2019), as presented in Figure 10. As can be seen from Figure 11c, NDVI was affected by drought, which indicated that the lower the vegetation coverage, the higher the drought. According to the results in Figure 11d, the response of the vegetation to drought was negatively correlated (about 97.0% of total pixels), of which negative correlation, about 3.25, 6.17, and 16.56% correlated as extremely (*p* < 0.01), moderately (*p* < 0.05), or weakly (*p* < 0.1) significant, respectively. Some studies have demonstrated the performance of NMDI_Soil_ for monitoring and predicting environmental risk conditions. The studies of Wang and Qu [19] indicated that integrating reflectance information from the near-infrared and short-wave infrared explained the drought severity. In addition, Wang et al. [8] reported that NMDI has a strong capability for detecting fires, and it pinpoints the active hotspots, as compared to other indices. Santos et al. [20] suggested the use of NMDI as the main drought index in future modeling.

## 4. Conclusions

The main goal of this study was to detect and monitor the spatiotemporal vegetation response to drought using Landsat 8 OLI images in the southeastern part of South Dakota, USA. The mean values of NDVI (from soybean, corn, and wheat in rotation) for 2016, 2017, 2018, 2019, 2020, and 2021 are 0.804, 0.757, 0.862, 0.843, 0.791, and 0.817, respectively, with a range of 0.276 to 0.956. However, in addition to environmental conditions, NDVI values also depend on the type of crop (in this study, soybean, corn, and wheat in rotation). The mean values of NMDI_Soil_ for the month of July in 2016, 2017, 2018, 2019, 2020, and 2021 are 0.513, 0.497, 0.474, 0.429, 0.289, and 0.299, with a range of 0.087 to 0.710. The results indicated that the relationships between vegetation and drought differed during the studied period (i.e., from 2016 to 2021). Based on the statistical criteria, vegetation coverage was negatively correlated to the drought in 69.4% of the total area for all of the studied years. It was about 97.0% of the total pixels when the highly correlated years (i.e., 2016, 2017, 2018, and 2019) were considered in this region. These results show that remote sensing data could be a powerful tool to monitor and detect vegetation coverage and response to drought, which could be helpful for regional vegetation protection.

## Figures and Tables

**Figure 1 sensors-23-02134-f001:**
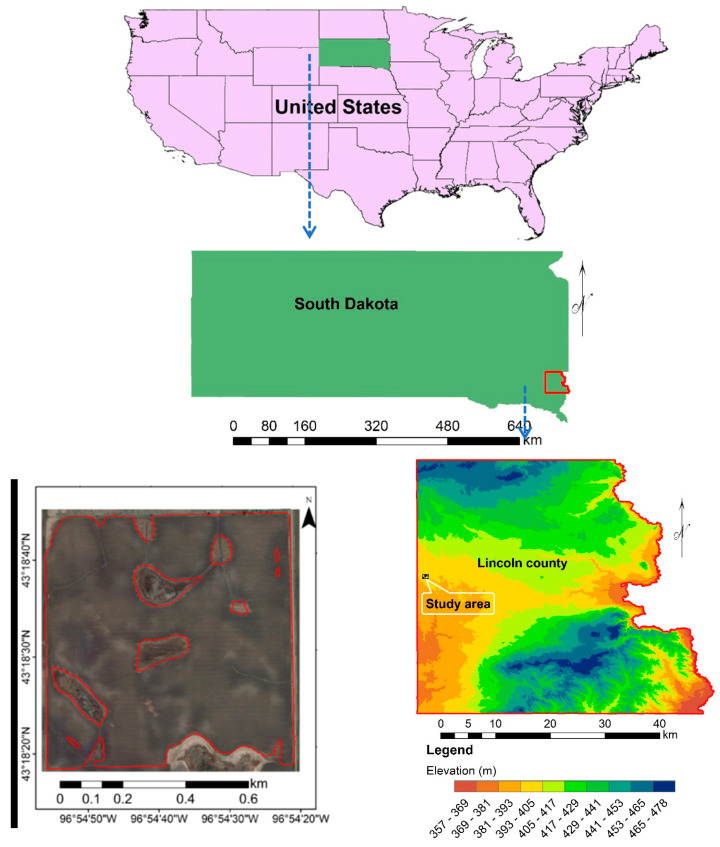
Location of study area in the United States.

**Figure 2 sensors-23-02134-f002:**
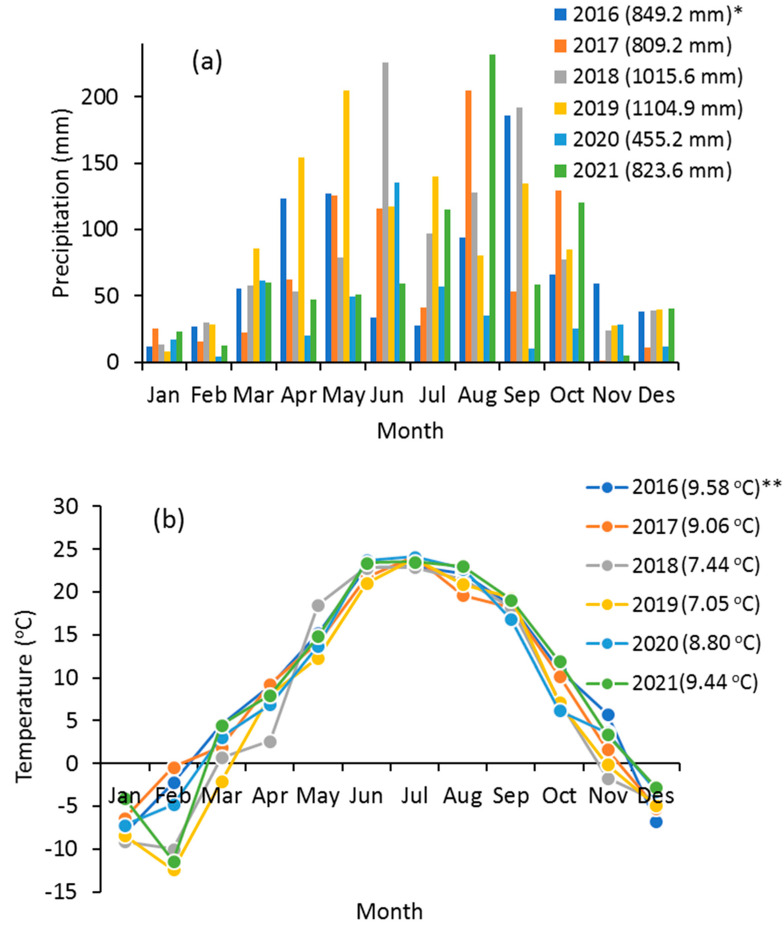
Average monthly precipitation and temperature from 2016 to 2021 (* cumulative precipitation; ** annual average temperature).

**Figure 3 sensors-23-02134-f003:**
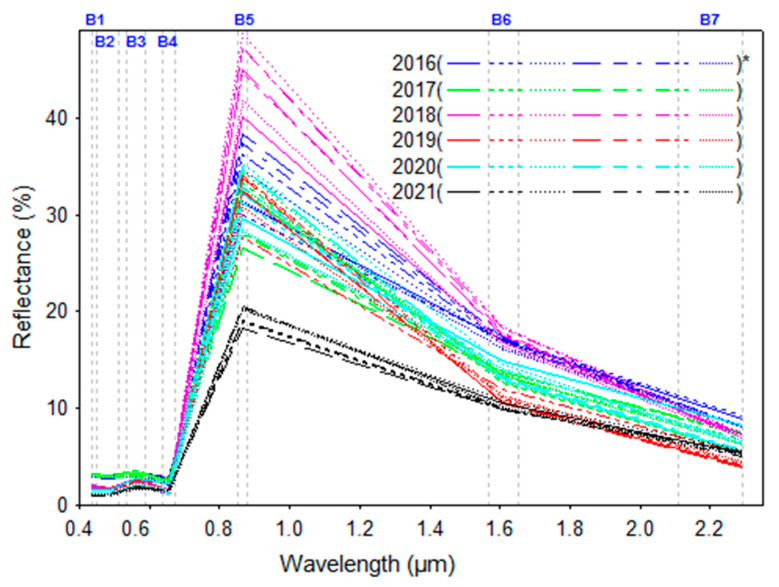
The spectral reflectance from some randomly selected pixels from the month of July in different years (* the selected pixels).

**Figure 4 sensors-23-02134-f004:**
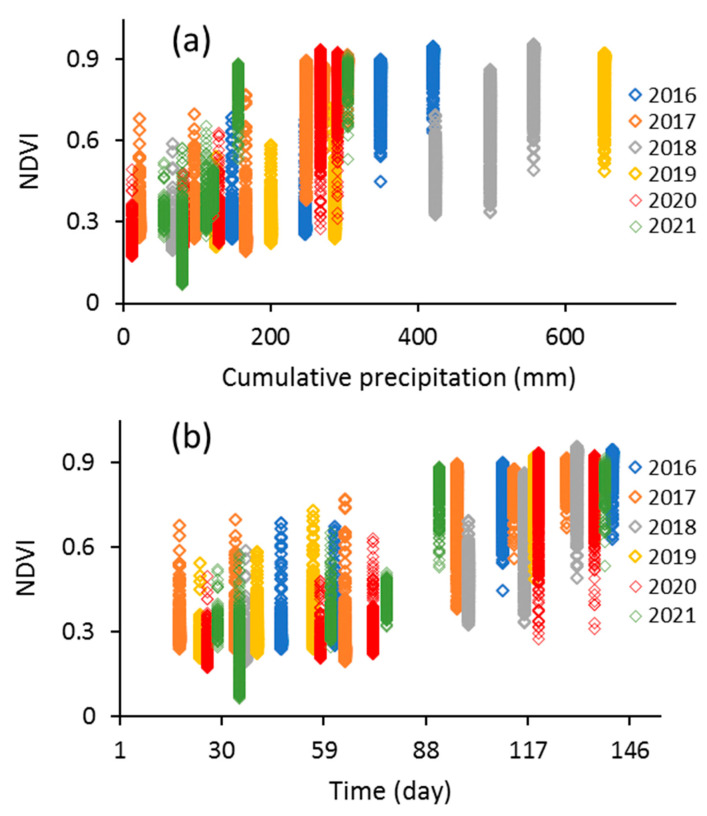
NDVI data (from soybean, corn, and wheat in rotation) vs. time (**a**) and precipitation (**b**) from April to August in different years at all pixels.

**Figure 5 sensors-23-02134-f005:**
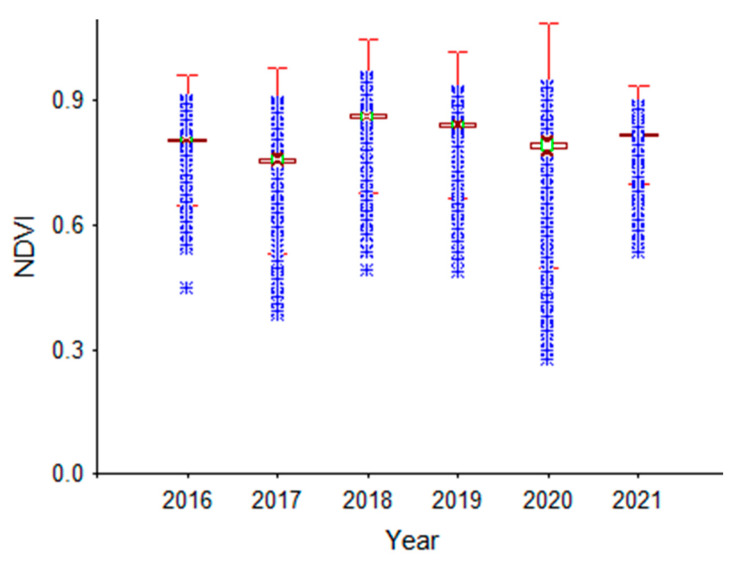
NDVI data (from soybean, corn, and wheat in rotation) for the month of July (2016–2021).

**Figure 6 sensors-23-02134-f006:**
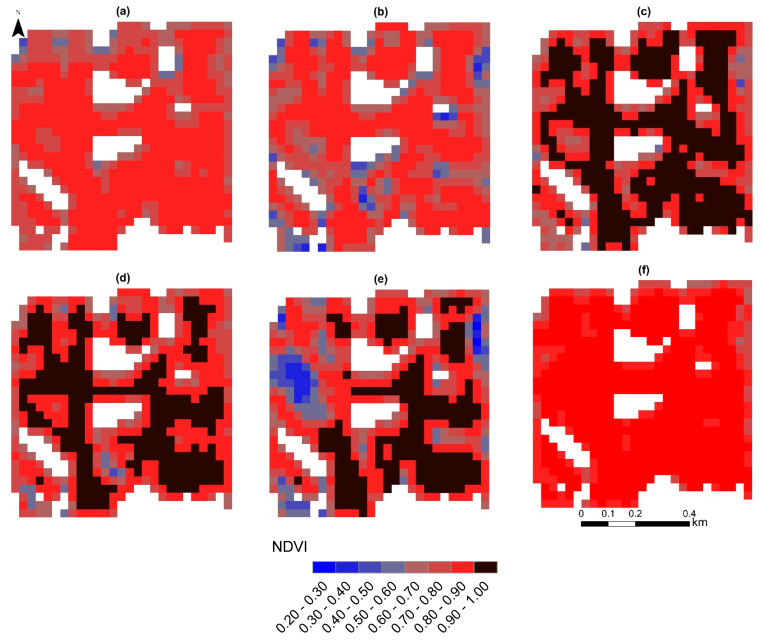
NDVI maps for the month of July: 2016 (**a**), 2017 (**b**), 2018 (**c**), 2019 (**d**), 2020 (**e**), and 2021 (**f**).

**Figure 7 sensors-23-02134-f007:**
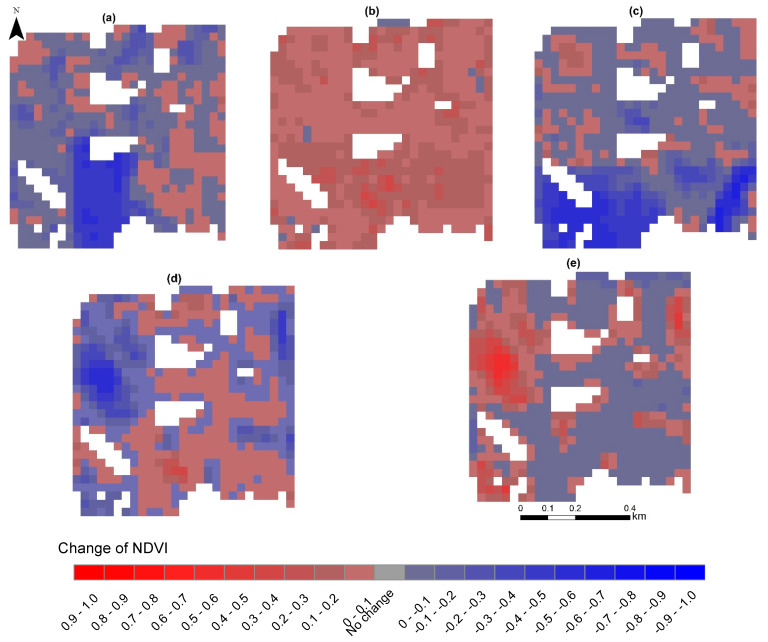
Change detection of NDVI maps for the months of July for 2016–2017 (**a**), 2017–2018 (**b**), 2018–2019 (**c**), 2019–2020 (**d**), and 2020–2021 (**e**).

**Figure 8 sensors-23-02134-f008:**
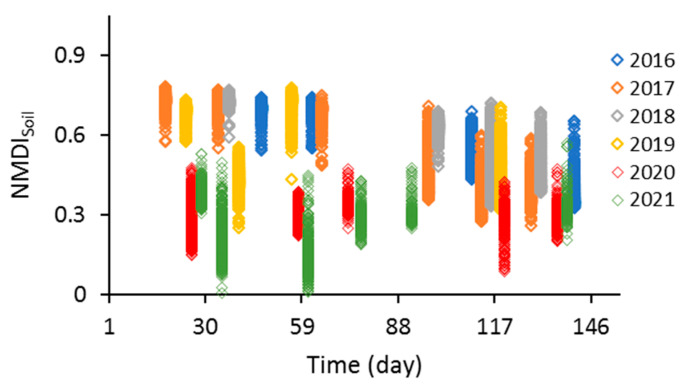
NMDI_Soil_ data vs. time for April–August in different years.

**Figure 9 sensors-23-02134-f009:**
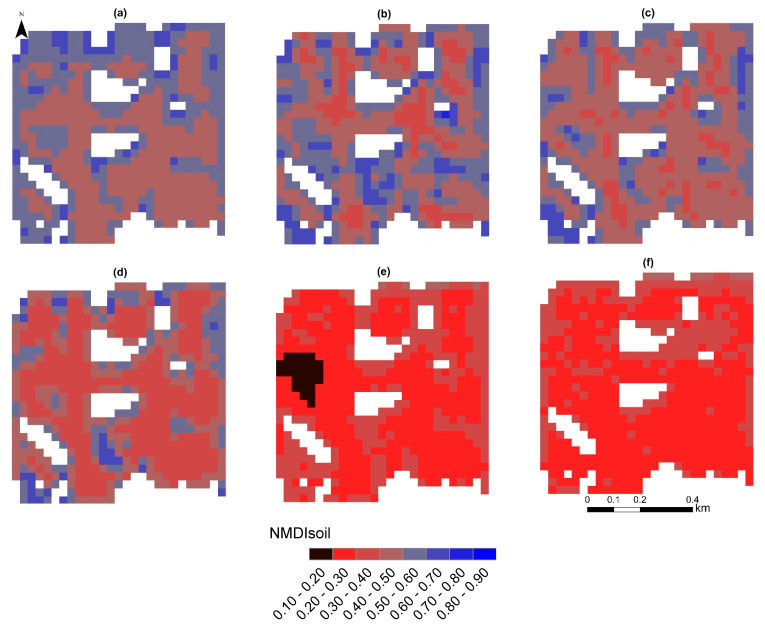
NMDISoil maps for the month of July for 2016 (**a**), 2017 (**b**), 2018 (**c**), 2019 (**d**), 2020 (**e**), and 2021 (**f**).

**Figure 10 sensors-23-02134-f010:**
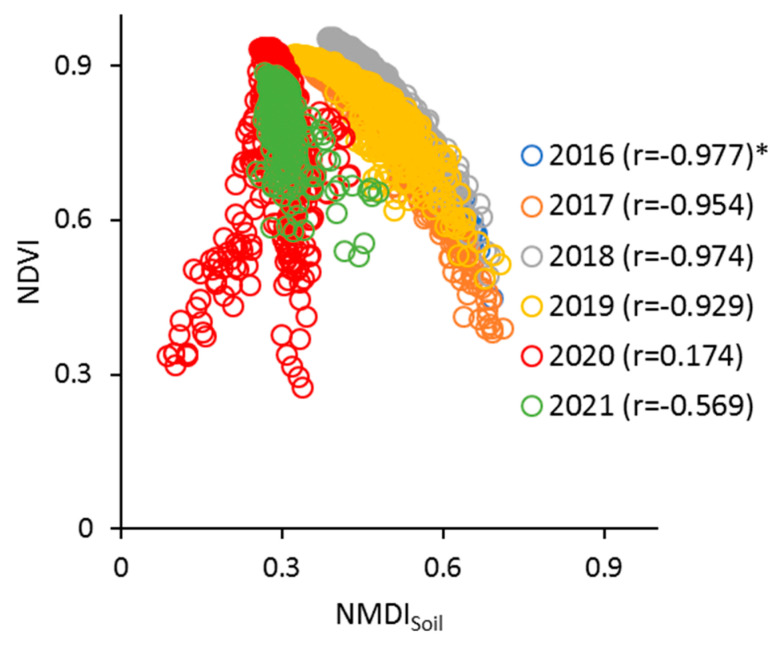
NMDI_Soil_ vs. NDVI data for the month of July (2016–2021) (* Pearson correlation coefficient).

**Figure 11 sensors-23-02134-f011:**
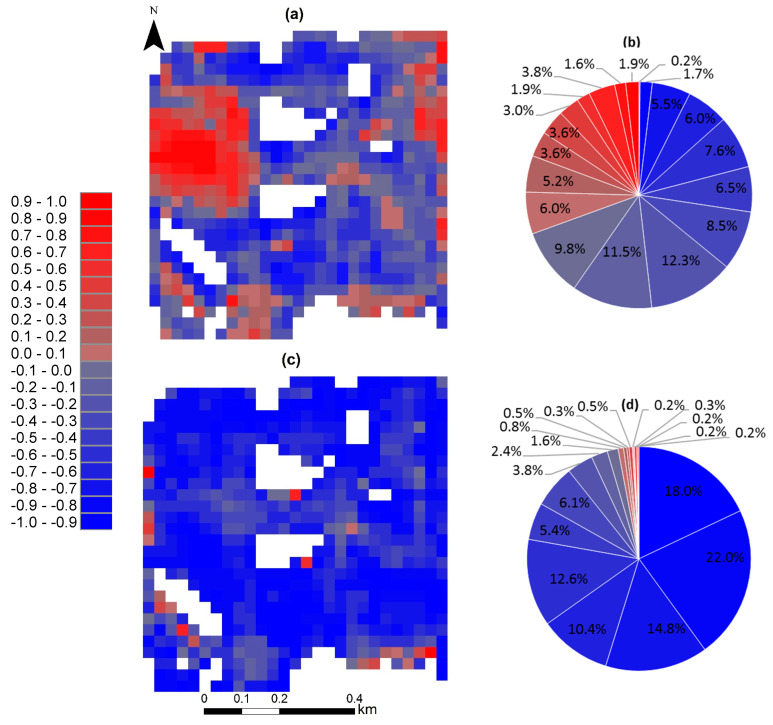
Pixel-by-pixel correlation between vegetation and drought (2016–2021 (**a**) and (2016–2019 (**c**)) and the percentage of pixels in the correlation classes (2016–2021 (**b**) and (2016 to 2019 (**d**)).

**Table 1 sensors-23-02134-t001:** Wavelengths and spatial resolutions for Landsat 8 OLI data.

Band	Abbreviation	Spectral Range (μm)	Spatial Resolution (m)
Coastal/Aerosol	B1	0.43–0.45	30
Blue	B2	0.45–0.51	30
Green	B3	0.53–0.59	30
Red	B4	0.64–0.67	30
NIR	B5	0.85–0.88	30
SWIR1	B6	1.57–1.65	30
SWIR2	B7	2.11–2.29	30

Abbreviations: NIR: near-infrared reflectance, SWIR: short-wave infrared reflectance.

**Table 2 sensors-23-02134-t002:** Descriptive statistics of NDVI data for the month of July (2016–2021).

Year	Min.	Max.	Mean	Median	Std. Dev	CV
2016	0.448	0.900	0.804	0.831	0.079	9.87
2017	0.382	0.896	0.757	0.793	0.112	14.82
2018	0.492	0.956	0.862	0.902	0.093	10.80
2019	0.488	0.922	0.843	0.885	0.088	10.45
2020	0.276	0.937	0.791	0.840	0.148	18.73
2021	0.531	0.887	0.817	0.839	0.059	7.20

**Table 3 sensors-23-02134-t003:** The results of change detection NDVI for the month of July, 2016–2021.

Change Type of NDVI	Amount of NDVI Change	2016–2017		2017–2018		2018–2019		2019–2020		2020–2021	
		Pixel No.	Area (%)	Pixel No.	Area (%)	Pixel No.	Area (%)	Pixel No.	Area (%)	Pixel No.	Area (%)
Increasing	0.9–1.0	-	-	-	-	-	-	-	-	-	-
	0.8–0.9	-	-	-	-	-	-	-	-	-	-
	0.7–0.8	-	-	-	-	-	-	-	-	-	-
	0.6–0.7	-	-	-	-	-	-	-	-	-	-
	0.5–0.6	-	-	-	-	-	-	-	-	5	0.79
	0.4–0.5	-	-	-	-	-	-	-	-	9	1.42
	0.3–0.4	-	-	4	0.63	-	-	2	0.31	18	2.83
	0.2–0.3	-	-	41	6.46	-	-	4	0.63	34	5.35
	0.1–0.2	1	0.16	244	38.43	19	2.99	16	2.52	73	11.50
	0.0–0.1	165	25.98	334	52.60	120	18.90	259	40.79	141	22.20
No change	0.0	-	-	-	-	-	-	-	-	-	-
Decreasing	0.0–−0.1	271	42.68	12	1.89	293	46.14	211	33.23	334	52.60
	−0.1–−0.2	80	12.6	-	-	43	6.77	73	11.50	21	3.31
	−0.2–−0.3	33	5.20	-	-	33	5.20	31	4.88	-	-
	−0.3–−0.4	13	2.05	-	-	33	5.20	19	2.99	-	-
	−0.4–−0.5	60	9.45	-	-	47	7.40	12	1.89	-	-
	−0.5–−0.6	12	1.89	-	-	41	6.46	8	1.26	-	-
	−0.6–−0.7	-	-	-	-	6	0.94	-	-	-	-
	−0.7–−0.8	-	-	-	-	-	-	-	-	-	-
	−0.8–−0.9	-	-	-	-	-	-	-	-	-	-
Total increasing	-	166	26.14	623	98.11	139	21.89	281	44.25	280	44.09
Total decreasing	-	469	73.86	12	1.89	496	78.11	354	55.75	355	55.91

**Table 4 sensors-23-02134-t004:** Descriptive statistics for NMDI_Soil_ data for the month of July (2016–2021).

Year	Min.	Max.	Mean	Median	Std. Dev	CV
2016	0.436	0.689	0.513	0.502	0.054	10.55
2017	0.356	0.710	0.497	0.490	0.081	16.34
2018	0.384	0.685	0.474	0.454	0.071	14.95
2019	0.318	0.706	0.429	0.411	0.083	19.38
2020	0.087	0.424	0.289	0.291	0.043	14.82
2021	0.252	0.477	0.299	0.295	0.024	8.11

**Table 5 sensors-23-02134-t005:** Regression model between NDVI and NMDI_Soil_ for the month of July (2016–2021).

Year	Regression Models	ME	RMSE	R^2^
2016	$\mathrm{NDVI}=-2.643{\left({\mathrm{NMDI}}_{\mathrm{Soil}}\right)}^{2}+1.377\left({\mathrm{NMDI}}_{\mathrm{Soil}}\right)+0.801$	−0.0004	0.014	0.967
2017	$\mathrm{NDVI}=-3.564{\left({\mathrm{NMDI}}_{\mathrm{Soil}}\right)}^{2}+2.327\left({\mathrm{NMDI}}_{\mathrm{Soil}}\right)+0.504$	−0.0002	0.023	0.957
2018	$\mathrm{NDVI}=-2.185{\left({\mathrm{NMDI}}_{\mathrm{Soil}}\right)}^{2}+0.916\left({\mathrm{NMDI}}_{\mathrm{Soil}}\right)+0.930$	−0.0001	0.018	0.962
2019	$\mathrm{NDVI}=-1.713{\left({\mathrm{NMDI}}_{\mathrm{Soil}}\right)}^{2}+0.595\left({\mathrm{NMDI}}_{\mathrm{Soil}}\right)+0.915$	−0.0004	0.029	0.884
2020	$\mathrm{NDVI}=0.603\left({\mathrm{NMDI}}_{\mathrm{Soil}}\right)+0.616$	−0.0004	0.145	0.031
2021	$\mathrm{NDVI}=-1.379\left({\mathrm{NMDI}}_{\mathrm{Soil}}\right)+1.229$	0.0002	0.048	0.322

## Data Availability

Not applicable.

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
