# Peer review of "Monitoring Spatiotemporal Vegetation Response to Drought Using Remote Sensing Data"

_sensors, 2023, doi:10.3390/s23042134_

Round 1

Reviewer 1 Report

1. The abstract needs to comprehensively introduce the methods applied in the manuscript;

2. Where all the data in Figure 2 come from, the source of the data should be given to help explain the validity and scientificity of the data;

3. The variables in all formulas in the manuscript should be explained, and the superscript, subscript, italic and bold of each variable should be unified. The format of multiple variables in the manuscript is not unified;

4. Lines 22-24, it is suggested to change "plant growth status" to "vegetation coverage";

5. The image quality is unqualified, and it is recommended to edit it again;

6. Figure 3 seems unnecessary. The text above the figure can already describe the image processing process clearly;

7. It is suggested to modify the title of 2.3, "Remote sensing data" cannot cover the contents of 2.3.1-2.3.3;

8. In lines 126-128, what does the change of reflectivity mean? It should be further explained.

Author Response

Dear Reviewer,

Thank you very much for reviewing the Manuscript ID sensors-2165465 entitled “Monitoring Spatiotemporal Vegetation Response to Drought Using Remote Sensing Data”. In reference to your comments and those of the reviewers, please find the revised version of our manuscript. All the suggestions and comments made by the reviewers have been considered and point-by-point responses made within the revised version shown by Track Change. Also, a separate file containing responses to the comments is provided and we believe has helped us improve it significantly. I look forward to receiving your reply regarding our revised manuscript soon.

Yours Sincerely

A. M. Nafchi

Reviewer 2 Report

A well-done job, but with tables and graphs of the same data. I suggest deleting one of them.

Author Response

(The authors gave the same response as above.)

Reviewer 3 Report

The manuscript entitled „Monitoring Spatiotemporal Vegetation Response to Drought Using Remote Sensing Data” presents interesting study on evaluation of Landsat 8 imagery for crop monitoring.

The manuscript is quite well prepared, however contains some drawbacks.

1) What Landsat 8 product was used for the analyses? Level 1 or level 2?

2) Please write equation 2 using bands (e.g. B5) like equation 1 to be clear what bands were used for calculation.

3) Please adjust alignment of the equations because the equation are out of the left margin.

4) Please add regression model which was used for the analyses in chapter 2.4. It would be better if this chapter has name “Statistical analysis”

5) What crop was cultivated in each year? Such information should be added in material and methods. Spectral reflectance is strongly dependent on crop and each crop have different spectral reflectance. Moreover growth stage of the crops during the acquisition of the imagery should be presented.

6) How the % of the reflectance presented in Fig. 4 was calculated? Usually raw data of the images have digital numbers not percentages of the reflectance. Please provide more details about the calculation of the % of reflectance.

7) Line 126: What criteria were used for random selection of the pixels? It was completely random or the pixels were from different location of the field?

8) Fig. 5: Please notice that NDVI is dependent on crop. If different crops were in different years it does not make sense to present the results in Fig. 5 for all years together (The same comment is valid for Fig. 6). What was the day=0 in Fig. 5b? Figures and tables should be self-explanatory, i.e. clear enough without reading all the manuscript.

9) Fig. 7 and 8: Why in the maps of NDVI are white areas? It is very low NDVI? Please explain. It would be good if one RGB map was presented to see the area of the field.

10) Did you verified if there is strong relationship between real soil moisture and NMDI? If not, such study in which relationship between two or more spectral indices is studied is of low importance. Usually various spectral indices are correlated but such information without further inference is not very useful.

11) The conclusions are very general because are not for certain crop. Most probably the results are crop dependent. The study should be more detailed and the conclusions should be more specific.

Author Response

(The authors gave the same response as above.)

Round 2

Reviewer 3 Report

The manuscript was corrected according all my comments. Current version pof the manuscript can be accepted for publication. Some technical changes are required, ir. in the references year should be in bold and names of the journals in italic font.

Author Response

Thank you very much for reviewing the Manuscript ID sensors-2165465 entitled “Monitoring Spatiotemporal Vegetation Response to Drought Using Remote Sensing Data”. In reference to your comments, please find the revised version of our manuscript. All the suggestions have been considered within the revised version and shown by Track Change.

Yours Sincerely

Ali M Nafchi